# Predictive Model of the Relationship between Positive and Negative Affect, Self-Consciousness of Appearance, and Positive Body Image in Physical Exercise Practice

**DOI:** 10.3390/healthcare12020187

**Published:** 2024-01-12

**Authors:** José Mendes, Pedro Alexandre-Sousa, Márcio Tavares

**Affiliations:** 1INTELECTO—Psychology & Research, 9500-373 Ponta Delgada, Portugal; 2School of Sciences and Humanities, University of Azores, 9500-321 Ponta Delgada, Portugal; 3Martingil Family Health Unit, 2419-014 Leiria, Portugal; pesousa@arscentro.min-saude.pt; 4Department of Nursing, Family and Community Health, School of Health, University of the Azores, 9500-321 Ponta Delgada, Portugal; marcio.fm.tavares@uac.pt

**Keywords:** positive body image, self-consciousness of appearance, satisfaction, positive affect, negative affect, physical exercise

## Abstract

The biopsychosocial development of individuals is influenced by body image and physical exercise. This study aimed to evaluate seven hypotheses regarding the impact of positive affect, negative affect, self-consciousness of appearance, and positive body image on the practice of physical exercise. The data were processed using the statistical package SmartPLS 4.0.9.5. Using an exploratory methodology based on structural equation modeling (SEM) of structural equations applied to small sample sizes, a tentative model has been generated to define the drivers of physical activity. The survey used the Body Mass Index, Body Appreciation Scale—Revised, and Derriford Appearance Scale—14 items. The study involved 129 men and 279 women (N = 408). The model demonstrated a distinct internal consistency in that, out of the seven hypotheses analyzed, only one was rejected. The findings of the multigroup analysis indicate that there are no statistically significant discrepancies between the constructed measures and the practice of physical activity. The conclusions of this study showed that, although positive and negative affect, and self-consciousness influence positive body image, there were no statistically significant differences between those who exercise and those who do not exercise. Various factors influence biopsychosocial development. Future studies should explore the influence of psychological and social variables on understanding body image and physical exercise.

## 1. Introduction

In recent decades, concern with appearance has been the object of great attention. Thompson [1] supports the relevance of understanding the influence of appearance psychology on individuals’ health and well-being. However, several factors influence body image satisfaction [2,3]. For example, Rounsefell et al. [4] maintains that dissatisfaction with body image may be due to the involvement of individuals in social networks, influencing food choices. A study by Durau et al. reveals that social networks, especially fitness influencers, may affect the increase of men’s and women’s physical activity [5].

Mendes et al. [6] analyzed dissatisfaction with body image and body mass index in a sample of male university students. The authors found that although the participants felt satisfied with their body, there was a relationship between BMI and fat mass perception. However, studies revealed that both overestimation and underestimation of weight and body image has influence on the tendency of less healthy behaviors [7,8,9]. These studies maintain that this type of behavior can lead the individual to resort to uncontrolled diets and excessive physical exercise. In fact, Sabastion et al. [10] reveals that most studies focused on understanding the relationship between physical activity and sports, other studies sought to explore how body image could be a protective or discouraging factor for physical exercise or sports participation. These authors maintain that participation in physical activity and sports was related to a less negative and more positive body image. However, it is important to underline that physical activity is appreciated as a fundamental element for a life to be considered healthy [11].

From another perspective, physical appearance has a relevant impact on social interactions and is estimated to be one of the characteristics most observed by others [12]. In this sense, physical activity can help to reduce a negative body image, and it is recommended to practice it at a frequency of 300 min a week or more [13,14]. Still, physical activity is not always used to establish a well-structured body image [15] due to the need for individuals to have a socially accepted body, generating dissatisfaction with appearance [6].

Byron-Daniel [16] maintains that there is ambiguity about the relationship between physical exercise and body image. However, the author mentions that “the exercise environment itself may also be crucial in encouraging, or discouraging, people with appearance motivations and concerns to become more active” (p. 301). Vani et al. point out that self-conscious emotions related to the body (e.g., shame, guilt, envy, and others) are highly prevalent in sports contexts. On the other hand, physical activity is an ideal intervention for body image and well-being [17,18]. Gori et al. support the existence of a mediation between body image concerns and addiction to physical exercise [19]. Thompson [1] contends that in the field of appearance research, a single model or theory is not sufficient. The studies conducted in this domain encompass a diverse range of factors, including but not limited to body image, attractiveness, disfigurement, and identity. Given that the relationship between positive and negative emotions, self-consciousness of appearance, and positive body image in the context of physical exercise involves multiple factors, Embodiment Theory was considered in the present study.

## 2. Positive Body Image, Self-Consciousness of Appearance, and Physical Exercise

Embodiment Theory is a multidisciplinary framework that examines the role of the body in shaping various aspects of human experience and cognition [20]. The Embodiment Theory proposes that our perception of the body is not solely a visual or cognitive process [21], but is intertwined with the individual’s bodily experiences [22]. They believe that a positive image involves appreciating the beauty of the body regardless of its culturally rooted ideals. The concept of body image holds significant importance and warrants examination within the context of physical activity [23]. The authors refer to physical activity as a type of lifestyle that includes structured exercise and sport. Physical activity is a fundamental element of a healthy life [11]. From a different perspective, it can be inferred that physical appearance has a significant impact on social interactions, and is regarded as one of the characteristics that are most frequently observed by others [12]. Studies have shown that physical exercise can reduce appearance concerns and promote a positive body image [24,25,26,27]. The way we feel inside our bodies influences the way we perceive our bodies [22,28].

An individual’s self-consciousness of appearance refers to their awareness and concern about their physical appearance [29,30]. For these authors, individuals can feel insecure about their appearance, which can influence their behavior, confidence, and social interactions. Consciousness of appearance is closely linked to bodily sensations and emotions, enabling body functionality. Body functionality has been equated with the domains of physical abilities and internal processes (such as muscle strength and physical endurance) [31]. From another perspective, the mental representation of the body also influences how the individual perceives themselves [32]. They are not solely cognitive, as bodily experiences influence them. Koban et al. [33] argues that mental and physical health are linked by neuronal systems that regulate physiology and cognition. The relationship between physical exercise and self-consciousness of appearance is complex and varies from person to person. However, it may require a holistic approach that includes psychological and emotional well-being to resolve the issues underlying self-consciousness of appearance [34].

Embodiment Theory posits that movement and physical activity encompass not solely the muscles and joints, but also all experiences involving the body and mind [35,36]. Physical exercise can have a positive effect on cognitive processes where the sense of achievement associated with physical activity contributes to a positive bodily experience [11,13].

The objective of this study is to understand the influence of body image on physical exercise. In this sense, we propose to test seven hypotheses and modeling to relate indicators and constructs to explain better the emotions (affects), the self-consciousness of appearance, and positive body image in physical activities.

**(H1)** 
*Positive affect has a positive impact on self-consciousness of appearance (DAS-14).*


**(H2)** 
*Negative affect has a positive impact on self-consciousness of appearance (DAS-14).*


**(H3)** 
*Positive affect has a positive impact on positive body image (BAS-2).*


**(H4)** 
*Negative affect has a positive impact on positive body image (BAS-2).*


**(H5)** 
*Affect positive have a positive impact on self-consciousness of appearance (DAS-14) which has a positive impact on positive body image (BAS-2).*


**(H6)** 
*Affect negative have a positive impact on self-consciousness of appearance (DAS-14) which has a positive impact on positive body image (BAS-2).*


**(H7)** 
*Positive body image (BAS-2) has a positive impact on physical exercise.*


## 3. Materials and Methods

### 3.1. Design

The design was quantitative and cross-sectional with random sampling.

### 3.2. Participants

A total of 408 individuals between the ages of 18 and 67 took part in this study (M = 34.42, SD = 10.93). Of all participants, 68.4% identified themselves as male and 31.6% as female; 68.6% practiced some kind of physical exercise/sport, of which 9.8% practiced exercise due to a doctor’s recommendation. Of the most prevalent forms of physical exercise, 15.7% were bodybuilding and personalized training, 12.7% were walking and running, 11.8% were choreographed group activities, 3.9% were non-choreographed group activities, and 3.9% were team sports games, among others. About the participants’ educational qualifications, 60.3% had higher education, 30.4% secondary, 8.3% 3rd cycle, and 1% completed 2nd cycle. When asked about their weight and height, 58.3% were of adequate weight, 26.2% were overweight, 8.1% had grade I obesity, 3.2% had grade II obesity, 2.9% were underweight, and 1.2% had grade III obesity. When asked how much they cared about their appearance, 51.5% said they cared a little about their appearance, 41.2% felt indifferent about their appearance, 4.2% cared too much, and 3.2% showed no concern about their appearance at all.

### 3.3. Data Collection and Measures

#### 3.3.1. Body Mass Index—BMI

It is an international classification that simply estimates whether an individual is underweight, normal weight, or overweight, using the present equation BMI = $\frac{Weight}{Height\;\times \;Height}$. Based upon this equation, individuals are classified as underweight when their BMI is less than 18.5; normal weight when it is between 18.5 and 24.9; pre-obesity between 25 and 29, and three types of obesity (class I, 30–34.9; class II, 35–39, and class III ≥ 40) [37,38].

#### 3.3.2. Body Appreciation Scale—Revised—BAS-2

The Body Appreciation Scale—Revised was developed by Tylka et al. [39] and validated for the Portuguese population by Marta-Simões et al. [40]. It assesses feelings and thoughts in relation to body image through a ten-item self-response scale consisting of items indicating 1 = Never, 2 = Rarely, 3 = Sometimes, 4 = Often, 5 = Always. Higher values indicate greater body appreciation, and the total score is calculated using the average of the 10 items. The original scale has a single-factor structure, excellent internal consistency, temporal stability after 3 weeks, and construct validity that is convergent and discriminant. The construct measured by BAS-2 is the same for both men and women, which allows us to make comparisons between these two groups. In this study, the scale has a good internal consistency index (Cronbach’s alpha = 0.93).

#### 3.3.3. Short-Form of the Portuguese Version of the Positive and Negative Affect Schedule—PANAS-VRP

It assesses specific emotional states, consisting of 10 emotions organized into 2 sub-scales: positive affect and negative affect. In the sum of the responses to the 5 positive and 5 negative emoticons (individually ranging from 1 to 5: 1 = “not at all or very slightly”, 2 = “a little”, 3 = “moderately”, 4 = “a lot”, 5 = “extremely”), the higher the score, the greater the affective experience (positive/negative) of the individuals [41,42]. In this study, the positive affect (Cronbach’s alpha = 0.86) and negative affect scale (Cronbach’s alpha = 0.87) have a good internal consistency index.

#### 3.3.4. Derriford Appearance Scale—14 Items (DAS-14)

It evaluates how the individual feels and behaves about the aspect that bothers them about their appearance, in which the statements are counted through two sets. Both sets of questions refer to the “characteristics of appearance”, and the second set refers to 14 statements of answers with five options (1—Not at all, 2—Slightly, 3—Moderately, and 4—Extremely, or from 0—N/A (not applicable); 1—Never/almost never, 2—Sometimes, 3—Often, and 4—Almost always) [29,43,44]. The presence of high scores indicates that the individual is experiencing stress and encountering challenges with their appearance. In this study, the scale has a good internal consistency index (Cronbach’s alpha = 0.91).

#### 3.3.5. Procedure

Before starting this study, an evaluation from the University of the Azores’ Ethics Committee was requested, which was positive (process number 15/2022). With the collaboration of two gyms in the city of Ponta Delgada (Portugal), the questionnaires and instruments were applied to their users anonymously. Also, to obtain a larger sample, the study was publicized via social networks (Facebook and Instagram), with a message stimulating general participation. The eligibility criteria comprised being over the age of eighteen, possessing the ability to read and write, and consenting to participate in the study. There were exclusion criteria, including not completing the data collection protocol. The contents explained the objective of the study and the participation mode, regardless of their physical activity. All ethical procedures were followed, including informed consent, free and spontaneous participation, and ensuring that the data collected were solely and exclusively used for this study. The participants completed a protocol comprising a sociodemographic questionnaire, which encompassed demographic information such as age, weight, height, gender, marital status, sports practice, educational qualifications, and preoccupation with physical appearance), the Body Appreciation Scale, and the Derriford Appearance Scale—short.

#### 3.3.6. Statistical Analysis

We used IBM SPSS for Macintosh, version 28 (Armonk, New York, NY, USA) to calculate several statistical analyses: Kaiser-Meyer-Olkin (KMO) Measure of Sampling Adequacy, Bartlett’s test of Sphericity and Cronbach’s alpha for the scales. This study was conducted using an exploratory methodology based on structural equation modeling with partial least squares estimation (PLS-SEM) using the SmartPLS 4.0.9.5 program, to assess how the study dimensions influence a positive body image [45], testing the respective hypotheses. Another multigroup partial least squares analysis was carried out to understand the differences in patterns of physical exercise practice [46]. Validity and reliability were assessed according to the reflective model [47], in which the indicators are interchangeable and strongly correlated [48].

## 4. Results

The assumptions regarding sample size, multivariate normality, and multicollinearity that are essential for this analysis have been thoroughly tested. All the instruments show homogeneity of the variables, which is considered excellent for the BAS-2 (KMO = 0.948, *X*^2^(45) = 2988.484; *p* < 0.001) and DAS-14 (KMO = 0.939, *X*^2^(91) = 2478.203; *p* < 0.001), and a good index for the PANAS-VPR (KMO = 0.849, *X*^2^(45) = 1968.247; *p* < 0.001).

After analyzing the estimated relationship in a reflective model and according to Carmines and Zeller [49], most of the outer loadings have values considered recommendable (λ ≥ 0.70), while six items in the DAS-14 and two items in the BAS-2 have lower outer loadings. Cronbach’s alpha coefficient (α) and the composite reliability (rho_A, rho_c) were used to assess reliability (Table 1).

The Heterotrait–Monotrait ratio (HTMT) is used to assess the absence of discriminatory validity, considering values below 0.90 [50,51]. Table 2 shows that all the constructs have adequate values.

Figure 1 shows the effect on endogenous variables, with strong correlations (0.5 ≤ |r| < 0.75) between the positive affect and BAS-2, negative affect and DAS-14, and DAS-14 and BAS-2 [52]. The endogenous variable BAS-2 (R^2^ = 0.510) is, predominantly, explained by the variable DAS-14 (R^2^ = 0.371).

Table 3 shows the direct effects and that the explained variance between the constructs is extremely significant for most of the constructs [53]. The highest percentage is attributed to DAS-14 → BAS-2 (31.8%).

The results can be supplemented with Figure 2, which relates the total effects (importance) to performance (average scores on a scale of 0 to 100). Here, it becomes clear that positive affect has an important total effect on positive body image.

Analyzing the latent variables of the structural model makes it possible to understand the relationship between the hypotheses in question. There are various tests to make the variable significant, such as *p*-value and the test *t*. Table 4 demonstrates that the relationship between the latent variables is significant (*p* ≤ 0.05) as a result of most of the hypotheses being validated, with only one hypothesis being rejected [54]. The statistical relationship between these constructs was significantly strong.

The standardized root mean square (SRMR) of the s residuals is considered suitable for studying discriminant validity, since values below 0.08 represent a good adjustment [50,55,56]. The results show that the SRMR is 0.059.

Henseler et al. [57] recommended the MICOM procedure (Measurement Invariance of Composite Models) consisting of three stages for evaluating the model: (i) configurational invariance (use of the same items in both models), (ii) compositional invariance (r = 1), and (iii) equal mean values and variances (<0.05). The first stage was valid due to the use of the same items in both models. The first stage was validated because the model in each group had the same number of constructs in both models. In the second stage, all the constructs were found to have perfect and almost perfect correlations [(BAS-2 = 1, negative affect = 1); DAS-14 = 0.998; positive affect = 0.995)]. After analyzing the mean values and equal variances, in the third stage, the total invariance would imply that both differences were equal to zero or that they were not significant. However, the results show that all the constructs have values greater than zero in which the permutation is extremely significant (<0.05) [57].

Table 5 analyzes the differences between users who exercise and those who do not. The results indicate that there is no direct relationship between the constructs. In this sense, there is no relationship between the constructs assessed and the practice of physical exercise.

Table 6 shows the coefficient of determination (R^2^), which reveals the correlation between the exogenous and endogenous constructs to explain the dependent variable. According to Chin [58], R^2^ values are significant (R^2^ = 0.67), moderate (R^2^ = 0.33), and weak (R^2^ = 0.19). BAS-2 as the dependent variable had an R^2^ = 51.4%, so the model presented has moderate explanatory power. According to the Stone–Geisser test (Q^2^), all endogenous constructs are satisfactory when Q^2^ > 0 [59,60]. Predictive validity is considered low, ranging from 0.02, moderate at 0.15, and high at 0.35 [55]. The results show that the DAS-14 and BAS-2 values indicate moderate predictive relevance.

## 5. Discussion

After analyzing the results, it is possible to observe that in the reflective model there are external loads of less than 0.70; although, in exploratory studies, external loads λ > 0.40 are accepted [48,61]. Once the model has been evaluated, all the constructs have an internal consistency of more than 0.70 [52,62]. O coefficient rho_A shows that all the constructs have a clear internal consistency. The rho_A coefficient shows that all the constructs have a clear internal consistency. In all cases, the composite reliability is higher than 0.70, which indicates that all the constructs have a clear internal consistency. The average variance extracted showed an acceptable value (AVE > 0.5), which demonstrates convergent validity in most of the constructs in the model, except DAS-14 [47,63].

After evaluating the effect of the endogenous variables, the results show that self-consciousness of appearance (DAS-14) is a fundamental construct in individuals’ perception of their body image. There was a high level of significance between the direct effects, in which positive and negative affect and self-consciousness of appearance seem to play a crucial role in the way individuals perceive their body image (H3, H4, H5, and H6). Thus, individuals tend to adjust appearance problems by understanding negative emotions (negative affect) through an avoidance behavioral response that often interrupts lifestyle, causing some distress [30,43].

Of all the hypotheses put forward, only the negative affect influencing self-consciousness of appearance hypothesis was rejected (H2). Body image is a person’s perception of their own body, including physical appearance, shape, size, and self-acceptance. These results may be because physical exercise is directly related to the level of subjective well-being and reduces negative emotions [30,43,64]. It should be noted that positive affect significantly influences self-awareness of appearance (H1). Gattario and Frisén [25] argue that individuals can improve their body image by implementing cognitive strategies and changing the social context in which they are inserted. These authors contend that interventions aimed at improving body image must not solely focus on physical appearance but also address individuals’ overall sense of belonging, agency, and empowerment.

The findings revealed that, despite being aware of one’s appearance and having a positive body image, there were no statistically significant differences in the outcomes of physical activity (Table 6). Although satisfaction with one’s body image influences consistency in physical exercise, the hypothesis presented was not supported (H7). The multigroup analysis showed no statistically significant differences between the constructs analyzed and the practice of physical exercise, corroborating other studies that also found no significant differences between exercisers and non-exercisers [65,66]. Understanding appearance-related problems enables the development of effective strategies for developing a more positive body image. Individuals who suffer from clinically significant levels of appearance self-consciousness have greater body image disturbance [43]. Baceviciene and Jankauskiene [26] argue that internalizing stereotypical notions regarding body fat can lead to body image concerns. Therefore, it is important to acquire strategies that reduce the internalization of stereotypical body ideals and promote a positive body image and physical activity. Body functionality is a valuable construct concerning positive body image and well-being [31]. Therefore, physical exercise should be implemented daily [64]. Engagement in physical exercise is a fundamental embodied practice, but exercise does not automatically foster positive embodiment simply by being in our bodies when we exercise [22].

## 6. Limitations and Future Studies

There are some important limitations of the present study that should be discussed. Firstly, the sample is not representative of the entire Portuguese population, since the major participants in the study live on an island in one of the Portuguese archipelagos. Secondly, the influence of social networks on physical exercise was not explored. In addition, other anthropometric data (e.g., fat mass, muscle mass, basal metabolism, others) were not explored. Finally, although the BAS-2 is applied to both female and male samples, it does not assess certain characteristics inherent to males (e.g., muscle mass). Future studies should focus on the influence of physical exercise on self-esteem, increased self-consciousness of appearance, social influence on beauty standards, and a focus on performance vs. aesthetics.

## 7. Conclusions

The relationship between a positive body image, self-consciousness of appearance, and physical exercise can be bidirectional and strongly influenced by individual, social, and cultural factors, so the results should be interpreted with care. In pursuit of the objective of this study, it is feasible to discern from the testing of the hypotheses that the constructs are not directly related to one other. However, it is imperative to promote a balanced approach towards these elements and foster a harmonious correlation between one’s physical appearance and their physical form. It is important to emphasize that the influence of exercise on body image can vary from person to person. Some people may have a negative perception of exercise, for example, if they practice it excessively or as a form of punishment for their body. It is, therefore, essential to approach physical exercise in a balanced and healthy manner. It is also important to remember that body image is a complex construction, influenced by psychological and social factors. Therefore, in addition to physical exercise, it is important to consider other ways of promoting mental health and well-being, such as seeking psychological support and practicing holistic self-care.

## Figures and Tables

**Figure 1 healthcare-12-00187-f001:**
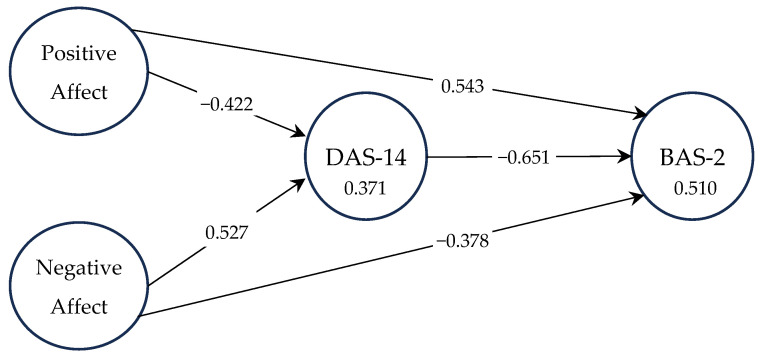
Coefficient of determination (R2) and correlations.

**Figure 2 healthcare-12-00187-f002:**
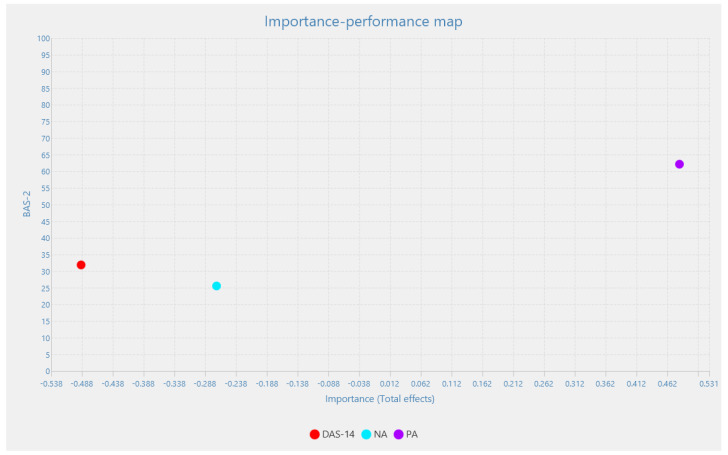
Priority map.

**Table 1 healthcare-12-00187-t001:** External load, reliability, and validity.

	Items	Loadings	Cronbach Alpha	Composite Reliability		Average Variance Extracted (AVE)
				Rho_A	Rho_c	
Positive Affect	Active	0.851	0.857	0.868	0.897	0.636
	Determined	0.830				
	Enthusiastic	0.754				
	Inspired	0.760				
	Interested	0.787				
Negative Affect	Upset	0.861	0.871	0.876	0.907	0.662
	Scared	0.860				
	Afraid	0.845				
	Guilty	0.759				
	Nervous	0.733				
DAS-14	Item1	0.787	0.913	0.921	0.925	0.471
	Item2	0.610				
	Item3	0.644				
	Item4	0.721				
	Item5	0.658				
	Item6	0.718				
	Item7	0.724				
	Item8	0.751				
	Item9	0.629				
	Item10	0.715				
	Item11	0.572				
	Item12	0.760				
	Item13	0.701				
	Item14	0.579				
BAS-2	Item1	0.630	0.938	0.952	0.948	0.647
	Item2	0.810				
	Item3	0.827				
	Item4	0.747				
	Item5	0.892				
	Item6	0.679				
	Item7	0.866				
	Item8	0.847				
	Item9	0.821				
	Item10	0.881				

Note: DAS-14 = Derriford Appearance Scale—14 items; BAS-2 = Body Appreciation Scale—Revised.

**Table 2 healthcare-12-00187-t002:** Fornell–Larcker discriminant validity criterion.

	BAS-2	DAS-14	NA	PA
BAS-2	0.80			
DAS-14	−0.655	0.687		
NA	−0.377	0.526	0.814	
PA	0.543	−0.423	−0.227	0.797

Note: BAS-2 = Body Appreciation Scale—Revised; DAS-14 = Derriford Appearance Scale—14 items; NA = negative affect; PA = positive affect.

**Table 3 healthcare-12-00187-t003:** Direct effect, explained variance and significance.

	Direct Effect	Correlations	Variance Explained	*p*-Value
Positive Affect → DAS-14	−0.319	−0.422	−0.135	<0.001
Negative Affect → DAS-14	0.455	0.527	0.240	0.254
Positive Affect → BAS-2	0.326	0.543	0.177	<0.001
Negative Affect → BAS-2	−0.047	−0.378	−0.018	<0.001
DAS-14 → BAS-2	−0.489	−0.651	−0.318	<0.001

Note: BAS-2 = Body Appreciation Scale—Revised; DAS-14 = Derriford Appearance Scale—14 items.

**Table 4 healthcare-12-00187-t004:** Hypothesis test (H1 a H6).

		Beta	2.5%	97.5%	t(|O/STDEV|)	*p*-Value	Supported
H1	PA → DAS-14	−0.319	−0.409	−0.228	6.867	<0.001	Yes
H2	NA → DAS-14	0.455	0.371	0.536	1.140	0.254	No
H3	PA → BAS-2	0.326	0.236	0.414	7.206	<0.001	Yes
H4	NA → BAS-2	−0.047	−0.125	0.034	10.900	<0.001	Yes
H5	PA → DAS-14 → BAS-2	0.156	0.104	0.214	5.525	<0.001	Yes
H6	NA → DAS-14 → BAS-2	−0.222	−0.285	−0.167	7.269	<0.001	Yes

Notes: BAS-2 = Body Appreciation Scale—Revised; DAS-14 = Derriford Appearance Scale—14 items; NA = negative affect; PA = positive affect.

**Table 5 healthcare-12-00187-t005:** Multigroup analysis of partial least squares between different types of users.

	Original Path Coefficient		Difference Path Coefficient	
	Physical Exercise (*n* = 243)	No Physical Exercise (*n* = 165)	Difference|(Total—Partial)|	*p*-Value
DAS-14 → BAS-2	−0.0743	−0.585	0.111	0.412
NA → BAS-2	0.027	−0.119	0.146	0.178
NA → DAS-14	0.472	0.572	−0.101	0.297
PA → BAS-2	0.398	0.243	0.155	0.187
PA → DAS-14	−0.368	−0.236	−0.132	0.253

Notes: BAS-2 = Body Appreciation Scale—Revised; DAS-14 = Derriford Appearance Scale—14 items; NA = negative affect; PA = positive affect.

**Table 6 healthcare-12-00187-t006:** Coefficient of determination (R^2^) and Stone–Geisser test (Q^2^).

Construct	R^2^	Q^2^
DAS-14	0.374	0.362
BAS-2	0.514	0.354

Note: BAS-2 = Body Appreciation Scale—Revised; DAS-14 = Derriford Appearance Scale—14 items.

## Data Availability

Data are contained within the article.

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
