# Peer review of "Predictive Model of the Relationship between Positive and Negative Affect, Self-Consciousness of Appearance, and Positive Body Image in Physical Exercise Practice"

_healthcare, 2024, doi:10.3390/healthcare12020187_

Round 1

Reviewer 1 Report

Comments and Suggestions for Authors

Thank you for the opportunity to review manuscript titled “investigating the positive and negative affect, self-consciousness of appearance, and positive body image in physical exercise practice.”

The authors assess the interaction between positive/negative emotions, self-consciousness, and positive body image. They test various hypotheses with reflective SEM modeling. However, the manuscript needs significant changes to help understand what authors are trying to convey.

Introduction

Adding more detail in the paragraphs will help understand how their study fits in the literature. Paragraph on body image needs to be expanded. As a reviewer, I was not sure why were above variables selected. Although they talk about physical exercise, it is unclear how they assessed physical activity in their participants? (#H7).

Methods

Which sample was selected? was it only gym attendees or general population from social media? If people from social media sites were recruited, then how was their physical activity evaluated? Similarly, how was BMI data collected? Was it a self-report from participants? What was the participant’s occupation.

Since the primary dependent variable is positive body image, so please consider expanding details about BAS-2 scale. Such details could be usefulness of the BAS-2 scale or how is it scored and what is the score range? Similarly, providing more details about the DAS-14 may help the readers.

Demographic data

It is unclear what sort of population is being considered. Did authors aim to capture all age ranges? Why wasn’t a specific group selected? (did any have a medical problem?) In their sample, very small percentage of patients were worried about their physical appearance so how does that explain or influence their results?

Results and discussion

Page 7, they reference there were not any differences between those who exercised and those who did not. But majority of their patients practiced some form of physical activity. It is unclear to me whether there is a novel finding in this analyses.

Discussion section could be enhanced.

Other limitations include recall bias IF participants self-reported above information.

Comments on the Quality of English Language

-

Author Response

Introduction

Adding more detail in the paragraphs will help understand how their study fits in the literature. Paragraph on body image needs to be expanded. As a reviewer, I was not sure why were above variables selected. Although they talk about physical exercise, it is unclear how they assessed physical activity in their participants? (#H7).

Answer: Thank you for your comments. A new section has been added to improve the description of the variables. The most practiced forms of physical exercise have been added.

Methods

Which sample was selected? was it only gym attendees or general population from social media? If people from social media sites were recruited, then how was their physical activity evaluated? Similarly, how was BMI data collected? Was it a self-report from participants? What was the participant’s occupation.

Answer: Thank you for your comments. These observations are described in the section 3.2 Data collection and Measures.

Since the primary dependent variable is positive body image, so please consider expanding details about BAS-2 scale. Such details could be usefulness of the BAS-2 scale or how is it scored and what is the score range? Similarly, providing more details about the DAS-14 may help the readers.

Answer: Thank you for your comments. More information about the instruments has been added.

Demographic data

It is unclear what sort of population is being considered. Did authors aim to capture all age ranges? Why wasn’t a specific group selected? (did any have a medical problem?) In their sample, very small percentage of patients were worried about their physical appearance so how does that explain or influence their results?

Answer: Thank you for your comments. These observations are described in the section 3.2.5 Demographic Data

Results and discussion

Page 7, they reference there were not any differences between those who exercised and those who did not. But majority of their patients practiced some form of physical activity. It is unclear to me whether there is a novel finding in this analyses.

Discussion section could be enhanced.

Other limitations include recall bias IF participants self-reported above information.

Answer: Thank you for your comments. More information about the instruments has been added.

Reviewer 2 Report

Comments and Suggestions for Authors

Good morning.

Firstly, I would like to congratulate the authors for their efforts in their study. They present interesting results. However, I consider that for the improvement of the final work and its acceptance, important improvements must be made.

The abstract is not clear. They should provide background information to justify their study. The results are also unclear and there are no conclusions.

The introduction is very limited, as the study is not well founded, taking into account the large volume of research that exists in this field of study.

They should carry out an in-depth reading of the whole text, in order to improve the translation and correct phrases and expressions. For example:

- Page 1. Lines 30-31: revise the sentence.

- Page 5. Line 144: “Positive Effect” and “Negative Effect”. Line 149: “DAS-14 e BAS-2”.

- Page 7. Line 189: Coefficient of determination (R2) e teste Stone-Geisser (Q2)

No information is provided on the measuring instruments. They should indicate their characteristics in order to justify their use, and clearly indicate what their measurement properties were in their adaptation and validation in the Portuguese context.

Demographic data. "When asked about their weight and height...", and "When asked how much they cared about their appearance". It is not mentioned whether an ad hoc questionnaire was used.

Lines 151-153. It is not clear where these data come from: "The total effects (importance) to performance (average scores on a scale of 0 to 100). No information is provided in the methods section.

Page 7. Demographic Data cited "A total of 408 individuals". In Table 4 two groups are mentioned "Physical exercise (n=433); No physical exercise (n=165)". The physical exercise group is higher than the one mentioned, while for the no physical exercise group it is not known where they come from and what their characteristics are. They should correct the data and provide all pertinent information.

The discussion is very brief and limited, considering the large number of variables and results. The discussion does not really exist.

Expand on the limitations of the study, as there are more than those mentioned.

They should provide the real conclusions of the study, not personal interpretations, as perhaps that should be left for the discussion.

Adjust references to the citation style of the journal

Author Response

The abstract is not clear. They should provide background information to justify their study. The results are also unclear and there are no conclusions.

Answer: Thank you for your comments. We believe that the abstract should present the study in a reduced form, encouraging readers to read it in full.

The introduction is very limited, as the study is not well founded, taking into account the large volume of research that exists in this field of study.

Answer: Thank you for your comments. A new section has been added to improve the description of the variables.

They should carry out an in-depth reading of the whole text, in order to improve the translation and correct phrases and expressions. For example:

- Page 1. Lines 30-31: revise the sentence.

- Page 5. Line 144: “Positive Effect” and “Negative Effect”. Line 149: “DAS-14 e BAS-2”. 

- Page 7. Line 189: Coefficient of determination (R2) e teste Stone-Geisser (Q2)

Answer: Thank you for your comments.

No information is provided on the measuring instruments. They should indicate their characteristics in order to justify their use, and clearly indicate what their measurement properties were in their adaptation and validation in the Portuguese context.

Answer: Thank you for your comments. More information about the instruments has been added.

Demographic data. "When asked about their weight and height...", and "When asked how much they cared about their appearance". It is not mentioned whether an ad hoc questionnaire was used.

Answer: Thank you for your comments. Information added in the section 3.2.1

Lines 151-153. It is not clear where these data come from: "The total effects (importance) to performance (average scores on a scale of 0 to 100). No information is provided in the methods section.

Answer: Thank you for your comments. These values are automated by the Smart PLS program.

Page 7. Demographic Data cited "A total of 408 individuals". In Table 4 two groups are mentioned "Physical exercise (n=433); No physical exercise (n=165)". The physical exercise group is higher than the one mentioned, while for the no physical exercise group it is not known where they come from and what their characteristics are. They should correct the data and provide all pertinent information.

Answer: Thank you for your comments. The figures have been corrected (transcription error).

The discussion is very brief and limited, considering the large number of variables and results. The discussion does not really exist.

Expand on the limitations of the study, as there are more than those mentioned.

They should provide the real conclusions of the study, not personal interpretations, as perhaps that should be left for the discussion.

Answer: Thank you for your comments. More information about the instruments has been added.

Reviewer 3 Report

Comments and Suggestions for Authors

The study examines a highly relevant topic in the field of psychology applied to behavior concerning sports practice, aiming for a healthy relationship between practitioners or athletes with their bodies and their cognition about it.

It is necessary to include the suggested modifications from the review report for the manuscript to be accepted for publication.

Author Response

Specific comments

Recommended Title: "Predictive model of the relationship between positive and negative affect, self-consciousness of appearance, and positive body image in physical exercise practice."

Answer: Thank you for your suggestion. We change the title.

Abstract

  • The conclusion described in the abstract is not based on the main conclusions derived from the obtained results.
  • It's suggested to modify the keywords to avoid being the same as those used in the manuscript title.

Introduction

  • This section should be supported by theories explaining behavior concerning the variables under study. For example: the theoretical paradigm explaining physical self- concept, motivation towards sports practice, or the theory explaining psychological constructs like positive and negative affect.
  • The novelty of this study concerning other studies analyzing physical appearance and affects descriptively and prospectively is not reflected.
  • Recent studies (last 3-4 years) of randomized controlled trials (RCTs) and/or quasi- experimental designs (with control groups and pre-post intervention designs) on the analyzed variables in the target population should be argued.
  • The study objectives are not provided. These should be written in infinitive before formulating hypotheses.

Answer: Thank you for your comments. A new section has been added to improve the description of the variables.

Methodology Research Design

  • The method should start with a description of the study's design, for example: "The design was quantitative, quasi-experimental, and cross-sectional (Hernández Sampieri, 2018)". The employed methodology (quantitative and/or qualitative) and the type of design according to the methodology should be specified (e.g., pre-post intervention design).
  • The research design needs to be described with the corresponding bibliographic reference.

Answer: Thank you for your comments. We have improved the information.

Participants
• Due to the age heterogeneity and origin of the sample, it's crucial to provide a table with sociodemographic characteristics as shown in this example:

Answer: Thank you for your comments. We have improved the data.

The text in section 2.2.5 "Demographic Data" corresponds to "Participants", not "Results".

  • The inclusion and exclusion criteria for participation in the study should be described. Measures or Instruments
  • Authors are advised to review the wording of the psychometric characteristics of the scales. Follow this example of description:

Motor Competence. The Scale of Perception of Children's Motor Competence (CMPI) by Ruiz and Graupera (2005) was used, assessing how young individuals perceive images depicting motor skills at two distinct resolution levels. This scale was chosen due to its design as a pictorial scale, even though the sample used was younger than the current one. It comprised 22 different measurements where participants indicated, using their finger or a pencil, the scene that best resembled what they felt or believed they could do themselves. All items were grouped into a single dimension, demonstrating an internal consistency of Cronbach alpha .87.

  • The indiscriminate use of the reliability measure, the same for all instruments, needs justification.

Answer: Thank you for your comments. We have performed the necessary review.

Procedure

  • Specific details about the message sent through social networks to recruit the sample

and how the participants were informed about the study's objective, as well as when

and how the scales were administered, are missing.

  • Why were social networks used to recruit the sample?
  • Was their level of sports practice considered? Were they sedentary individuals, regular physical activity practitioners, or athletes?

Answer: Thank you for your comments. We have performed the necessary review and added information.

Data Analysis

  • It's crucial to mention that non-parametric inferential statistical tests were used due to

the non-normal distribution of the sample.

  • The Kolmogorov-Smirnov test is not provided. Concerning this test, assuming the mean

and population variance as known, which in most cases is impossible, makes the test very conservative and less powerful. To address this issue, a modification of the Kolmogorov-Smirnov test known as the Lilliefors test was developed. The Lilliefors test assumes that the mean and variance are unknown, being specially developed to test normality.

Answer: Thank you for your comments. We have performed the necessary review and added information. We used the KMO to validate the sample size, multivariate normality, and multicollinearity.

Results

  • Table 2 would correspond to mean, standard deviation, and bivariate correlations data

(not provided) of the study variables. For example:

Important note regarding the Results

A structural regression analysis is conducted with a measurement model and a structural equation model that are not described in the study. It is recommended to review this article:

  • Doral Fábregas, F.; Rodríguez Ardura, I.; Meseguer Artola, A. Models of structural equations in social science research: User experience in Facebook. Rev. Cienc. Soc. (Ve) 2018, 24(1), 22-40. https://doi.org/[insert DOI number if available].

Discussion

- It is necessary to report the confirmation or refutation of the hypotheses raised and compare the results with current studies (last 3-4 years) on the study variables.

and biased nature of the sample treatment, the characteristics of the control group, etc.

Answer: Thank you for your comments. We have performed the necessary review and added information.

Conclusions

The conclusions must be directly related to the objectives, which should be written in the Introduction.

Answer: Thank you for your comments. We have performed the necessary review and added information.

Round 2

Reviewer 1 Report

Comments and Suggestions for Authors

Revised version is much better. 

Author Response

Thank you for your comments, which have done much to improve our work.

Reviewer 3 Report

Comments and Suggestions for Authors

The Participants section should be placed after section 3.1. as section 3.2. Participants.

The correct order of Method and Materials is detailed:

- Research Design

- Participants. The sampling technique and the inclusion and exclusion criteria should be included.

- Variables and instruments

- Procedure

- Data analysis

Author Response

Thank you for your comments, which have done much to improve our work. We have reorganized the information as requested.